# Similarity Index Values in Fuzzy Logic and the Support Vector Machine Method Applied to the Identification of Changes in Movement Patterns During Biceps-Curl Weight-Lifting Exercise

**DOI:** 10.3390/jfmk10010084

**Published:** 2025-02-28

**Authors:** André B. Peres, Tiago A. F. Almeida, Danilo A. Massini, Anderson G. Macedo, Mário C. Espada, Ricardo A. M. Robalo, Rafael Oliveira, João P. Brito, Dalton M. Pessôa Filho

**Affiliations:** 1Instituto Federal de Educação, Ciência e Tecnologia de São Paulo (IFSP), Piracicaba 13414-155, SP, Brazil; andreperes@ifsp.edu.br; 2Graduate Programme in Human Development and Technologies, São Paulo State University (UNESP), Rio Claro 13506-900, SP, Brazil; tiagofalmeida.w@gmail.com (T.A.F.A.); dmassini@hotmail.com (D.A.M.); andersongmacedo@yahoo.com.br (A.G.M.); 3Department of Physical Education, School of Sciences (FC), São Paulo State University (UNESP), Bauru 17033-360, SP, Brazil; 4Post-Graduation Program in Rehabilitation Sciences, Institute of Motricity Sciences, Federal University of Alfenas (UNIFAL), Alfenas 37133-840, MG, Brazil; 5Instituto Politécnico de Setúbal, Escola Superior de Educação (CIEQV—Setúbal), 2914-504 Setúbal, Portugal; mario.espada@ese.ips.pt (M.C.E.); ricardo.robalo@ese.ips.pt (R.A.M.R.); 6Sport Physical Activity and Health Research & INnovation CenTer (SPRINT), 2040-413 Rio Maior, Portugal; 7Centre for the Study of Human Performance (CIPER), Faculdade de Motricidade Humana, Universidade de Lisboa, 1499-002 Cruz Quebrada, Portugal; 8Comprehensive Health Research Centre (CHRC), Universidade de Évora, 7004-516 Évora, Portugal; 9School of Sport, Santarém Polytechnic University, Av. Dr. Mário Soares, 2040-413 Rio Maior, Portugal; rafaeloliveira@esdrm.ipsantarem.pt (R.O.); jbrito@esdrm.ipsantarem.pt (J.P.B.); 10Research Centre in Sport Sciences, Health Sciences and Human Development (CIDESD), Santarém Polytechnic University, 2040-413 Rio Maior, Portugal

**Keywords:** pattern recognition, motor activity, theoretical models, resistance training

## Abstract

**Background/Objectives**: Correct supervision during the performance of resistance exercises is imperative to the correct execution of these exercises. This study presents a proposal for the use of Morisita–Horn similarity indices in modelling with machine learning methods to identify changes in positional sequence patterns during the biceps-curl weight-lifting exercise with a barbell. The models used are based on the fuzzy logic (FL) and support vector machine (SVM) methods. **Methods**: Ten male volunteers (age: 26 ± 4.9 years, height: 177 ± 8.0 cm, body weight: 86 ± 16 kg) performed a standing barbell bicep curl with additional weights. A smartphone was used to record their movements in the sagittal plane, providing information about joint positions and changes in the sequential position of the bar during each lifting attempt. Maximum absolute deviations of movement amplitudes were calculated for each execution. **Results:** A variance analysis revealed significant deviations (*p* < 0.002) in vertical displacement between the standard execution and execution with a load of 50% of the subject’s body weight. Experts with over thirty years of experience in resistance-exercise evaluation evaluated the exercises, and their results showed an agreement of over 70% with the results of the ANOVA. The similarity indices, absolute deviations, and expert evaluations were used for modelling in both the FL system and the SVM. The root mean square error and R-squared results for the FL system (R^2^ = 0.92, r = 0.96) were superior to those of the SVM (R^2^ = 0.81, r = 0.79). **Conclusions**: The use of FL in modelling emerges as a promising approach with which to support the assessment of movement patterns. Its applications range from automated detection of errors in exercise execution to enhancing motor performance in athletes.

## 1. Introduction

Monitoring the movement of the barbell is common in weightlifting exercises to assess the performance of the practitioner [1]. In the bicep-curl exercise, supervision of the barbell movement by a professional tends to prevent injuries resulting from improper loads, postures, and executions [2].

The supervision of resistance exercises by a personal trainer plays an important role in preventing injury and in promoting the physical development of practitioners. The involvement of an experienced professional ensures that exercises are performed with proper technique, with postures and loads adjusted as necessary. This personalized attention not only minimizes the risk of injuries that can occur due to improper execution but also maximizes the benefits of training [3].

One of the main obstacles to adhering to supervised physical exercise is the high cost of personal training services. Studies show that hiring a professional for individualized support can be financially unfeasible for a large portion of the population, leading many to abandon regular exercise programs [4].

With the advancement of technology, fitness apps have become increasingly popular as alternatives to personal training services. These apps offer a variety of features, including personalized workout plans and progress tracking, at a lower cost [5].

However, the accuracy of these apps has been questioned. Several studies indicate that the accuracy of data collected by digital devices can vary significantly, raising uncertainties about their reliability compared to professional supervision. Research suggests that many users may not achieve the expected results due to errors in algorithms or in the interpretation of the collected information [5].

In summary, while digital apps provide an accessible solution for monitoring physical exercise, the costs associated with professional supervision and uncertainties regarding the accuracy of emerging technologies continue to be significant barriers to effective adherence. The combination of these factors suggests an urgent need to develop solutions that integrate the best of both worlds: professional support and digital technologies.

Increasing the load on the bar during bicep-curl exercises can change movement patterns. Werner et al. [6] investigated how different loads (60%, 85%, and 95% of an athlete’s one-repetition maximum—1 RM) influence movement patterns. The results indicated that relative load has a significant effect on movement patterns. This suggests that as the load increases, athletes may adopt different movement patterns, which is relevant for training, especially for younger athletes who are perfecting their technique. Furthermore, the study highlighted that while lighter loads may allow for more repetitions, this could lead to the risk of developing movement patterns that are not ideal for maximum loads.

The trajectories of joint movements during exercise execution can serve as a basis for the analysis of movement patterns. These analyses help in identifying aberrant movement behaviours and evaluating sports technique. The goal is to provide quantitative and reliable feedback on movement quality, assisting coaches in making more accurate decisions regarding athlete performance and safety. Additionally, they can be used to investigate the relationships between movement patterns and injury risk [7].

Generally, motion data describe movement trajectories, each consisting of a temporal sequence of recorded locations for an object. In this work, we utilize the trajectory of a marker on the bar to analyse its movement trajectory [8]. Analysing human movement trajectories as time series is not uncommon and is part of current research [9].

Despite technological advancements, form analysis during practice of resistance exercises remains a rare feature of monitoring apps. There are some available apps that perform video motion analysis, such as OpenCap, Mirror AR, Runmatic, Spark Motion, Onform, and PhysioMaster. However, this functionality, coupled with immediate feedback at the end of each execution, could bring enormous benefits for training safety and effectiveness. Automatic analysis of exercise form, for example, would greatly reduce the risk of injuries and could enable feedback to be provided after each repetition of a resistance exercise [10].

To implement this feature, it is necessary to employ mathematical methods for movement analysis. The trajectory of a movement, as previously mentioned, can be analysed as a time series that can be compared to a good movement reference and analysed for needed alterations. To this end, a similarity test can be employed to identify pattern changes between different executions of an exercise and a standard model established according to the guidelines and protocols of the National Strength and Conditioning Association (NSCA) [11].

The Morisita–Horn (MH) dispersion or overlap index [12] is a measure of similarity or difference between two data sets. The index ranges from 0 (no similarity) to 1 (complete similarity). It is a measure used in ecology to quantify the overlap between two communities or species samples. It is particularly useful for assessing the similarity in species composition between different habitats or populations, allowing for an understanding of how species share resources or occupy similar niches [12]. In the present study, this index was adapted for the analysis of two bar-movement trajectories during the execution of the biceps-curl weight-lifting exercise.

The values obtained for the MH index for correct executions, within an acceptable range of variation, can be taken as a reference for comparison with executions involving additional load; these data can then be subjected to an analysis of possible changes. However, to identify changes in exercise execution patterns (changes in spatial trajectory), a mathematical model based on fuzzy logic (FL) has shown good results [13]. In this case, the values obtained with the MH index were used to establish degrees of membership and rule sets for the FL system.

FL has demonstrated good performance in modelling human thought processes and is capable of handling uncertainty [14]. It can be designed to simulate how humans think and make decisions, especially in situations where information is imprecise or incomplete [15]. This is particularly relevant in healthcare and also in physical training, where assessments often rely on subjective interpretations by the professional making the diagnosis [16]. As the values of the MH index vary on a scale from 0 to 1, a regression model based on machine learning called the Support Vector Machine (SVM) model was also applied to quantify the accuracy of movement, and its performance was compared with that of the model obtained using FL.

SVM is a machine learning algorithm used for classification and regression that seeks to find the optimal hyperplane that separates different classes of data. It can be applied to classify and recognize movement patterns, such as physical activities or gestures [17], aiding in areas such as rehabilitation [18], sports [19], and human−computer interaction [20]. Additionally, it has a significant advantage in movement analysis: its ability to handle high-dimensional data and robustness against overfitting (high accuracy on training data but poor prediction on unseen data) makes it effective for analysing complex movement data [21,22].

In this study, we aimed to analyse the suitability of the MH index for training a machine learning procedure to identify movement-pattern alterations during a weight-lifting exercise. To design the model, a common single-joint exercise (biceps curl with a barbell) was modelled based on the hypothesis that deviations from the Cartesian coordinates of the bar position in the sagittal plane (e.g., horizontal and vertical displacements) that may reflect significant pattern alterations due to load increase can be automatically detected using FL analysis associated with the MH scale. Moreover, the deviations in displacement measured through traditional means (such as absolute deviations) can be related to similarity changes detected by the MH index to quantify the degree of similarity between executions using SVM algorithms. In addition, variance analyses between groups (ANOVA) and human evaluators might increase the confidence associated with using similarity indices in modelling with FL and SVM regression, resulting in a viable approach to employing mathematical models in automated movement analysis.

## 2. Materials and Methods

### 2.1. Participants

Twelve male volunteers, all with more than six months of experience in resistance training, participated in the study. At the beginning of the tests, the volunteers self-reported their training sessions, with 8 to 10 maximum repetitions, 3 to 4 sets per exercise, and 6 to 9 exercises per session, 3 to 5 times per week. Two volunteers were excluded because they were unable to complete all the repetitions. The remaining ten (age: 26 ± 4.9 years, height: 177 ± 8.0 cm, body weight: 86 ± 16 kg) completed all the proposed repetitions. The study was approved by the ethics committee of the local university (protocol: 17486119.0.0000.5398).

### 2.2. Procedures

Data collection was conducted in the Laboratory of Human Sports Performance Optimization (LABOREH). Participants used a green semi-spherical marker measuring 25 mm in diameter that was fixed to the barbell.

The volunteers performed a sequence of three complete repetitions of a biceps-curl weight-lifting exercise using only the barbell (9 kg/considered no load). After a ten minute rest, they performed three more repetitions of the exercise with a load (using the barbell with weights) of 25% of their body weight. This was followed by another break and three additional repetitions of the biceps-curl weight-lifting exercise with a load of 50% of their body weight [8]. Volunteers were instructed to perform each set with a similar cadence and were told this cadence should be close to that used in their daily routine to avoid unusual performance conditions. Additional guidelines for the performance of the biceps curl were as follows: (i) the inter-hand distance was measured during the first set and maintained throughout the attempts; (ii) the exercise was performed with a full range of movement (with ascending and descending phases), using an external focus; (iii) the participants were instructed to avoid sagittal oscillations of the trunk and barbell, any movement or impulse of lower limbs, and exaggerated elevation of the scapulae; and (iv) the technique was controlled by the researchers via feedback for the participant when a correct technique was observed [23]:

For the collection of temporal positional data during resistance exercises, a digital video camera attached to a Galaxy S9 smartphone (Samsung^®^, Suwon-si, Gyeonggi-do, Korea) with 12 megapixels and UHD 4K resolution was used. The camera was stationary, with its optical axis perpendicular to the participant’s sagittal plane, as shown in Figure 1 [8].

The procedure for calibrating the measurements was based on distances between markers in the background (forming a right triangle) and in a plane coincident with the participant’s sagittal plane. Additionally, actual measurements of some body segments of the volunteers (upper arm, forearm) and their actual heights were used to verify the measurements. This approach made it possible to determine displacement measurements from the two-dimensional coordinates of the participant in the plane they occupied [8,9].

Video capture in MPEG-4 format was conducted for three complete executions of the proposed exercises for each of the three load variations. The recordings were made at a frequency of 30 frames per second [24,25], and the video acquisition time corresponded to three executions of the respective exercise for each load [8].

Digital processing of the videos was performed using Wondershare Filmora version 9 (Wondershare Filmora, Hong Kong, China) [26] to apply a Chroma Key effect [27] on the colour of the markers and to apply an Alpha channel [28] to better contrast the markers with the rest of the environment in the scene. Kinovea 0.8.27 software (Kinovea, Bordeaux, France) [29] was used for semi-automatic tracking of the markers, and their coordinates were exported to Extensible Markup Language files. The origin of the Cartesian coordinate system was assigned to the centre of the marker located on the bar at the beginning of the upward movement [8].

### 2.3. Obtaining Displacement Measurements

Displacement measurements were taken from the marker located on the bar and calculated from its movement in relation to the *x*-axis, as follows:(1)$$∆x=x_{f}-x_{i}$$
where Δ*x* is the displacement, *x_f_* is the x-value of the coordinate at the end point, and *x_i_* is the x-value of the coordinate at the origin.

Displacement in relation to the *y*-axis was calculated as follows:(2)$$∆y=y_{f}-y_{i}$$
where Δ*y* is the displacement, *y_f_* is the y-value of the coordinate at the endpoint, and *y_i_* is the y-value of the coordinate at the origin.

### 2.4. Human Evaluators

To obtain a qualitative analysis for comparisons that reflects what is common in physical assessments, two specialists with over 30 years of experience (experts in evaluation) visually analysed the performances of the exercises executed with different load variations. They observed the exercises performed by the volunteers through videos, and their data were used as an observational reference for statistical/mathematical models. They used as a reference the execution performed with only the bar, without added load. When they observed the executions performed with two different loads, they attempted to identify any changes in horizontal and vertical displacements versus the reference exercise execution (i.e., the silhouette of motion for the repetitions with no load added) [9]. For this observational analysis, the observer recorded the part of the silhouette that differed from the reference motion The results of the observer analysis were further tested by ANOVA (see section below) to determine whether the standard deviation from the reference motion coordinate was significant and therefore corroborated the results of the observer’s analysis by the observer; these results were then used to determine whether the alteration in movement patterns should be incorporated into the movement modelling using FL.

### 2.5. Statistical Analysis

The descriptive statistics, normality tests, and analysis of variance of the data were obtained using SPSS^®^ version 22.0.0 (SPSS, Corp, Armonk, NY, USA) [30]. A one-way ANOVA was conducted to verify the existence of significant differences in the maximum displacement (vertical and horizontal) during the lifting phase of the exercise as the average of three full-range joint repetitions with each load. Tukey’s post-hoc test was applied with a significance level of *p* < 0.05. Using the maximum values of absolute deviations of the displacements, a one-way ANOVA was conducted with Tukey’s post-hoc test at a significance level of *p* < 0.05 to verify the existence of significant differences in displacement among the three different loads.

Morisita–Horn Index

The MH index was calculated using Equation (3), which was presented by Horn [12], as follows:(3)$$I_{MH}=\frac{2\sum_{i=1}^{S}x_{i}y_{i}}{\left(\frac{\sum_{i=1}^{S}x_{i}^{2}}{X^{2}}+\frac{\sum_{i=1}^{S}y_{i}^{2}}{Y^{2}}\right)XY}$$
where *x_i_* is the abundance of species *i* in sample *x; y_i_* is the abundance of species *i* in sample *y*; and *X* and *Y* are the number of species for the samples. The equation was implemented using the average of the maximum displacement values from the three no-load executions (with only the bar) as a reference and compared to the maximum values obtained in the other executions. The no-load executions (training for correct execution) Ex1, Ex2, and Ex3 generated the average value of maximum displacements, MMD0. This average value was compared to the other executions, here designated as follows: Ex1_25, Ex2_25, Ex3_25 for the load of 25% of the subject’s body weight and Ex1_50, Ex2_50, and Ex3_50 for the load of 50% of the subject’s body weight. The absolute and relative deviation values in relation to the mean were also obtained.

The results obtained from ANOVA, which showed significant differences in displacement, along with the analysis by the specialists, provided the criteria for modelling marker positioning data using FL based on the MH indices and absolute deviations found. The chosen model for identification was Sugeno with 18 inference rules, considering the maximum absolute deviation obtained via computer vision and the load used (0%, 25%, or 50%). The FL system, which was implemented using the Fuzzy Logic Designer app, employed the weighted average defuzzification method.

Since the values of the MH index present a similarity scale, regression was also performed using SVM. The results obtained from ANOVA, which showed significant differences in displacement, were also utilized. The entire regression process conducted by SVM was developed using the Regression Learner app, comparing the following models: linear SVM, quadratic SVM, cubic SVM, fine Gaussian SVM, medium Gaussian SVM, and coarse Gaussian SVM. In all cases, cross-validation for five groups was used to protect against overfitting. Pearson correlation calculations, polynomial fit, Morisita–Horn indices, fuzzy modelling, and SVM modelling were all performed using Matlab^®^ 24.1.0 software (Matlab, Portola Valley, CA, USA) [31].

## 3. Results

The averages of the maximum displacement values of the bar during the upward phase of the exercise for all participants were calculated in centimetres. For a load of 0%, using only the bar, the average horizontal displacement was 23.7 ± 5.9 cm. For a load of 25%, it was 23.6 ± 5.0 cm, and for a load of 50%, it was 24.9 ± 4.3 cm. The average vertical displacement was 56.8 ± 7.7 cm for a load of 0%, 59.6 ± 8.5 cm for a load of 25%, and 64.0 ± 9.0 cm for a load of 50%.

The maximum values of absolute deviations (in cm) obtained in each execution with loads of 25% (Ex1_25, Ex2_25, and Ex3_25) and 50% (Ex1_50, Ex2_50, and Ex3_50), using the average of the maximums from the first three executions (MMD0) as a reference, are displayed in Table 1 and Table 2.

In a comparison of the average maximum horizontal displacement values from the first three executions (using only the bar) with the executions at 25% and 50% of the subject’s body weight, no significant difference was found within the group. The same comparisons for vertical displacement showed a significant difference within the group only when the displacement values with the no-load exercises were compared to displacement values with the repetitions at 50% of the subject’s body weight, yielding a *p*-value of *p* = 0.002.

Thus, MH indices were calculated for the maximum vertical displacements for each volunteer for each execution/load. Table 3 displays these index values with comparisons between MMD0 and the maximum values from executions performed at 50% load.

The human evaluators conducted a qualitative analysis of the exercises recorded on video with respect to the load variations. Using as a reference the exercise performed without added load (with only the bar), they were able to perceive variations only when the no-load executions were compared with those performed at 50% of the subject’s body weight. When the assessments by Ev.01 and Ev.02 of the difference between individual attempts were compared to the ANOVA results, the following results were obtained: ANOVA and Ev.01 showed 70% agreement, and ANOVA and Ev.02 showed 80% agreement.

As a result, only the values of deviations and MH indices in the comparison between MMD0 and 50% load were used to create the mathematical models. Figure 2 displays the graph of MH values and absolute deviations with a trendline given by the quadratic equation, as follows:(4)$$y=-3.1026\times 10^{-5}x^{2}-4.1535\times 10^{-4}x+1.0013$$
with an R-squared value of 0.92 and a Pearson correlation coefficient of 0.96.

Based on the evaluations of the specialists, a range of absolute deviation values was established for the identification of statistically significant changes in movement patterns as a result of increased load. These deviation values, along with the MH index values, served as the basis for modelling the FL system. A deviation range was established within the closed interval of [0, 25], with zero representing no error and 25 representing the maximum deviation value relative to the mean of the initial position. Figure 3 displays how the system was created: two inputs, the maximum deviation value from the execution and the load value (the system was assigned load variation even though no significant difference was observed by ANOVA), were used. The variation in deviation was subdivided into six parts using Gaussian functions (Figure 4).

The output values were modelled with the MH variation within a range of 97 to 100% (0.97, 1.0) similarity between the comparisons. The values obtained from the FL system and the original values are displayed in Figure 5. The Pearson correlation coefficient for the two sets of values was 0.96, and R-squared = 0.92.

To compare the methods, a regression model using SVM was also employed. The model was trained with the absolute deviation values and MH indices for the comparisons of MMD0 with the executions at 50% load. The following models were trained: linear SVM, quadratic SVM, cubic SVM, fine Gaussian SVM, medium Gaussian SVM, and coarse Gaussian SVM. The root mean square error (RMSE) and R-squared values are displayed in Table 4.

As shown in Table 4, the best SVM model was the coarse Gaussian. With this model, we obtained the comparison between the original values and those obtained from the model (Figure 6). The Pearson correlation coefficient for the two sets of values was 0.79.

## 4. Discussion

This study identified disturbances in movement due to increased load during the execution of a barbell bicep-curl exercise using a similarity index. The use of maximum absolute deviation for the analyses is justified by the need to identify significant changes in movement patterns, with the no-load execution serving as the reference. The maximum absolute deviation values were used to verify the existence of significant differences in range of motion with the different loads utilized, an approach similar to that used in the work by Peres et al. [8]. This allows for an assessment of an individual’s ability to lift a certain load while utilizing muscular contraction to overcome resistance and control sources of joint instability and thereby ensuring safety and effectiveness in training.

In the biceps-curl weight-lifting exercise, the primary movement is elbow flexion, which primarily involves the biceps brachii muscle. When weight is added, the force required to perform the movement increases, resulting in a change in vertical displacement during the exercise; vertical displacement is thus more sensitive than horizontal displacement to load variations [32]. In contrast, horizontal movement involves rotational movements that are not the primary focus of the bicep curl. During this exercise, the involved joints, especially the elbow, do not perform significant movements [8]. The bicep curl is predominantly a uniaxial movement focused on elbow flexion and extension and does not involve the lateral or rotational displacements that would be necessary to generate significant differences in horizontal displacement [23].

Only the comparisons that showed significant differences in displacement were subjected to similarity analysis. The use of the similarity index allowed for the comparison of two executions of the bar movement as time series. In this study, the MH index was used to compare the movement performed with only the bar to the movements executed with added load. The index ranges in value from 0 to 1, where 0 indicates no similarity and 1 indicates complete similarity [12]. A high index value suggests that the maximum deviations of the movements are very similar among the analysed individuals, indicating consistent technical execution. Lower values may indicate significant variations in technique or movement execution, which could be relevant for adjustments in training or for developing personalized rehabilitation or performance programs. The analysis using the MH index provided a quantitative approach to identifying changes in movement patterns due to load variations, contributing to a better understanding of the variables that influence practitioner performance [33]. Table 3 displays the similarity values of the MH index. The variation of the index ranged from 0.97582 to 0.99998, encompassing absolute deviation variations within the range of 0.53 to 24.43 cm. This result reflects that the execution movements are similar, but the load variations identified as significantly altering the movement fall within this range.

The human evaluators, when analysing movement with reference to the execution performed without added load, were able to observe variations only for executions with a load of 50% of the subject’s body weight, a result corroborated by the ANOVA results. The percentage agreement between ANOVA and evaluators for responses identifying or not identifying significant displacements relative to the established reference (no load) and other executions (with load) varied between 70% and 80%, while agreement among evaluators was equal to 90%. In the comparison of the results obtained from ANOVA and human evaluators, there was good agreement both between the evaluators and the model and among evaluators in some cases. Good agreement among evaluators indicates consistency in error perception in human assessment [9,34]. However, evaluating exercise quality via human judgment is subjective and time-consuming [35].

The results obtained from the MH indices, along with the values of maximum absolute deviations, are displayed in Figure 2 in a scatterplot with a trendline described by a quadratic function generated by Matlab^®^ version 24.1.0 [31], which showed an R-squared value of 0.92, within acceptable limits for model fitting. This fit would already serve to relate observed values of absolute motion deviations with significance index values of MH; however, as this work aimed to incorporate characteristics of human experts into movement assessment, FL was used for modelling and relating load, vertical absolute deviation, and similarity index, which was taken to indicate movement disturbance.

With the MH variation values ranging from 0.97582 to 0.99998 and absolute deviation variations within the range of 0.53 to 24.43 cm, a comparison of results from ANOVA and human evaluators established a similarity-index variation of 97 to 100% (0.97, 1.0) for identifying changes in movement patterns, with maximum absolute deviation values between 0 and 25 cm, where zero indicates no error and 25 represents the maximum deviation relative to the mean of the initial position.

The FL system was created using the Sugeno inference method, relating the maximum absolute deviation values from each execution for different loads (0%, 25%, or 50%), even though significant differences (as measured with similarity index values) were observed only when the 50% load variation was compared with the 0% load variation. These similarity index values indicate, within the presented scale, the accuracy of the movement compared to the no-load reference. This variation can be associated with the degree of accuracy in performing the movement described in Figure 4, which includes six execution levels (very low, low, low medium, medium, medium high, and high), represented in the system by Gaussian functions. In this case, these levels allow for feedback at the end of the upward execution of the exercise, where the feedback is based on classifying the executed movement.

Siow, Chin, and Kubota [35] similarly proposed an FL system to evaluate simple exercises based on human skeleton poses. Their goal was to provide a more objective and efficient way to assess exercise quality for older adults. The researchers’ approach consisted of four steps: (1) converting video into a sequence of human skeleton poses; (2) extracting essential poses from the sequence; (3) generating FL membership functions based on pose similarity and exercise completeness; and (4) performing rule inference and defuzzification to obtain an exercise score. The method was evaluated across four types of execution: (1) correct exercise, (2) incorrect pose, (3) incomplete sequence, and (4) different exercise. In our work, we opted to classify performance into five categories: (1) Total Error, (2) Medium Error, (3) Average Execution, (4) Medium Accuracy, and (5) Total Accuracy. The authors concluded that their method could distinguish between different types of exercise executions and provide a reasonable exercise score. They concluded that this method presents a promising technique for evaluating simple exercises using human skeleton poses.

Work by Huang et al. [36] also proposed evaluating rehabilitation exercises through joint tracking in an intelligent system based on FL that was capable of tracking and quantifying joint-movement effectiveness in patients. The evaluation system was developed to address inaccuracies and uncertainties in joint movements. The inputs from joint movements were transformed into degrees of membership by projecting numerical input values into a set of membership functions using FL set inference.

After results were obtained with the FL system and compared with polynomial fitting performed earlier, an additional test was conducted to compare different machine learning methods: one based on evaluations from experts in resistance-exercise assessment (fuzzy) and one based solely on similarity related to maximum absolute deviation (using the same values for fitting presented in Figure 2). Thus, a regression model using SVM was employed. The RMSE and R-squared results displayed in Table 4 show that the linear and coarse Gaussian models yielded the best results. The coarse Gaussian model yielded an R-squared value of 0.81, which was superior to that of the linear model (0.72). The Pearson correlation coefficient was also calculated for the data obtained from the coarse Gaussian model and the observed data from original MH indices, yielding a value of 0.79.

A comparison of the initial quadratic fit with the FL model and the SVM models reveals that in terms of R-squared and Pearson correlation coefficients, both quadratic fit and FL models outperform SVM models. Both quadratic fit and FL models yielded R-squared values equal to 0.92 and correlation coefficients equal to 0.96, while the best SVM yielded an R-squared of 0.81 and a correlation coefficient of 0.79.

FL is particularly effective in dealing with uncertainties and variabilities in data. Unlike polynomial fitting, which assumes a precise and deterministic relationship between variables, as noted by Fouzia, Khenfer, and Boukezzoula [37], FL modelling allows for working with imprecise or vaguely defined information, better reflecting the complexity of human joint movements, which can be influenced by multiple nonlinear factors and complex interactions.

FL modelling offers a more flexible framework for capturing nonlinear relationships. These models can be adjusted for different conditions or contexts without the need to rewrite the underlying mathematical function. This is especially useful in biomedical applications, where data characteristics can vary significantly among individuals or situations [38].

The FL model, particularly the Takagi−Sugeno model, is considered a universal approximator and has proven to be very accurate in modelling nonlinear systems, often outperforming polynomial models in terms of accuracy and interpretability. Furthermore, the flexibility of FL models allows for better adaptation to uncertainties and nonlinearities compared to traditional polynomial models. Therefore, in many cases, the FL model may be more advantageous [39].

Regarding FL modelling and SVM, although it has already been shown that there are differences in RMSE and R-squared values that favour the FL model in this case, there are still additional advantages to the FL model over SVM for modelling human movement data. According to Apostolopoulos and colleagues [40], many practical applications related to the control and modelling of dynamic systems, such as human movements, often see FL-based systems outperform SVM. This is due to the ability of FL models to handle uncertainties and model complex relationships between variables, which is crucial in dynamic and nonlinear contexts like those associated with human movements, as corroborated by the work of Andújar and Zaitseva [38,39].

FL-based systems are particularly effective at capturing interactions and dependencies among different factors that influence human behaviour, allowing for more intuitive and interpretable modelling. Additionally, they can be easily adapted to include new information or variables, which is an advantage in dynamic environments where conditions can change rapidly. In contrast, while SVMs are powerful in classification and regression tasks, they may not be as effective at modelling complex dynamic relationships without significant pre-processing and careful feature selection [40]. Therefore, in contexts where modelling dynamic systems and interpreting relationships between variables are essential, FL models may be a superior choice compared to SVMs.

Besides this, the current study did not aim to design a universal reference movement for a biceps curl from which the model could recognize imperfections in performance by anyone (e.g., skilled or non-skilled subjects, subjects with disabilities) or dysfunctional executions. The findings apply only to the alteration of position from the beginning to the end of an exercise with a load increment. For example, the ascendent (lift) phase was considered essential in functional terms, since it is based on the ability of the contractile mechanisms of muscle to produce concentric torque; the muscles must be able to overcome resistance and perform the exercise with control of the degrees of instability (i.e., keeping load and body parts within the trajectory expected for the movement), therefore ensuring optimal performance and safety [41]. In addition, since the analysis of human movement by human observation is based on large-scale features, with a low level of detail (i.e., spatial accuracy and temporal resolution), the size of an alteration of movement that could be detected by visual examination is dependent on a priori knowledge of the type of movement or the viewpoint from which the movement is observed [42]. Moreover, advanced motor performance might show functional variability and that an experienced observer might judge these functional particularities to be qualitatively different [43]. Therefore, the support of an automated analysis applying a machine learning algorithm can improve correspondence among judges, assisting them in judging a movement when no body model is available. To that end, a direct relationship between execution and the reference motion must be established [44,45].

One limitation of the present method lies in the fact that it evaluates movement displacement in a two-dimensional (2D) manner. In contrast, three-dimensional (3D) analyses provide a more comprehensive view of movement across different planes, allowing for the assessment of changes in joint position that may occur to maintain proper posture under increased load. Moreover, a 3D model including the sagittal plane would provide more accurate data by avoiding parallax and perspective errors often associated with 2D analyses [8]. It is important to note, however, that the barbell bicep curl predominantly occurs in the sagittal plane rather than in the transverse and frontal planes. While the 3D approach presents advantages over 2D analysis, two-dimensional analyses like those conducted in this study still allow for movement evaluation using data obtained from any type of video recording. Other limitations are the small sample size, small number of evaluators, and low temporal recording resolution (30 Hz). Regarding the temporal resolution, it is important to note that the movement cadence was not high, so the 30 Hz video recording was probably sufficient to avoid loss of temporal positional data during the biceps curl. In addition, it is important to consider that the influence of frame resolution on automatized image analysis remains to be addressed. Finally, the expert visual analysis of variations in the trajectory of the movement may be a source of systematized error, since the level of correspondence between the different trajectories is associated with the accuracy of the cognitive representation of the movement pattern to be analysed [46]. Therefore, additional information on the movement to be analysed, such as might be collected via a third evaluator, detailed kinematics, and inertial information could contribute further insights on the features to be observed, hence improving confidence in the human analysis [42,44,47].

Future research should consider analysing individuals’ movements across different planes beyond the sagittal plane using three-dimensional coordinates (3D). This approach will enable a more detailed investigation into the positional sequence of joints and objects while allowing for an expansion of the FL system beyond the results obtained through two-dimensional analysis. Furthermore, future studies using human expertise for visual examination of movement should try to minimize the sources of errors by assisting the observation process with the following: (i) more one independent evaluator, since alignment between three evaluators trends to increase confidence in the analysis; (ii) detailed kinematics information (e.g., relative and absolute angle-to-angle curves) regarding limb position and joint coordinate during the movement; and (iii) the inertial information (e.g., 3-D vectors of limb or barbell temporal and spatial parameters, as measured using accelerometer, gyroscope, and magnetometer signals) to quantitatively interpret the causes of movement alterations. Thus, modelling using FL presents itself as a promising tool to assist in diagnosing motor patterns, with applications ranging from identifying errors in movement execution during exercise to optimizing motor performance in athletes.

## 5. Conclusions

The results of the present study indicate that the MH similarity index can be utilized to identify changes in movement patterns due to increased load during the execution of the biceps-curl weight-lifting exercise. Furthermore, the indices provide sufficient information for modelling in automated exercise evaluation systems. It was observed that variations due to increased load in the horizontal axis are not significant for the biceps-curl weight-lifting exercise. The comparison between two machine learning models, FL and SVM, demonstrated the superiority of FL modelling in applications involving human movement. The analysis conducted here showed that an FL system can provide more accurate information about movement patterns than human visual observation. Since the work was carried out using standard video recordings on a smartphone, this highlights that the use of the FL model can be easily implemented on mobile devices, emphasizing the practicality of automated supervision for monitoring movement using smartphone cameras.

## Figures and Tables

**Figure 1 jfmk-10-00084-f001:**
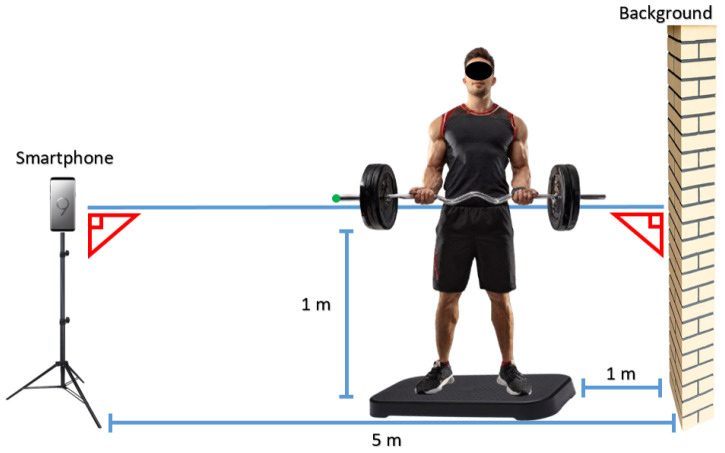
Scheme for capturing videos. Image partly generated by AI in: https://firefly.adobe.com/generate/images (accessed on 14 September 2024).

**Figure 2 jfmk-10-00084-f002:**
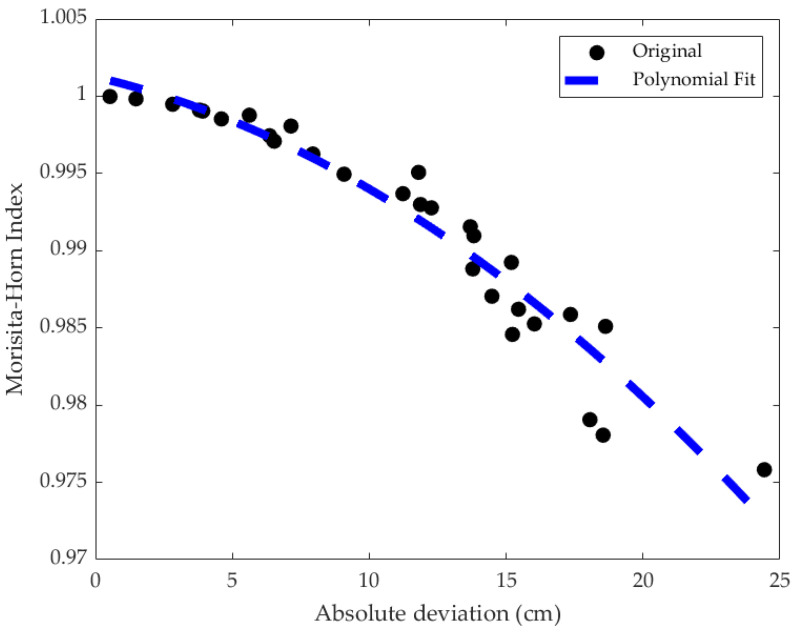
Scatterplot of MH index values vs. absolute deviation values.

**Figure 3 jfmk-10-00084-f003:**
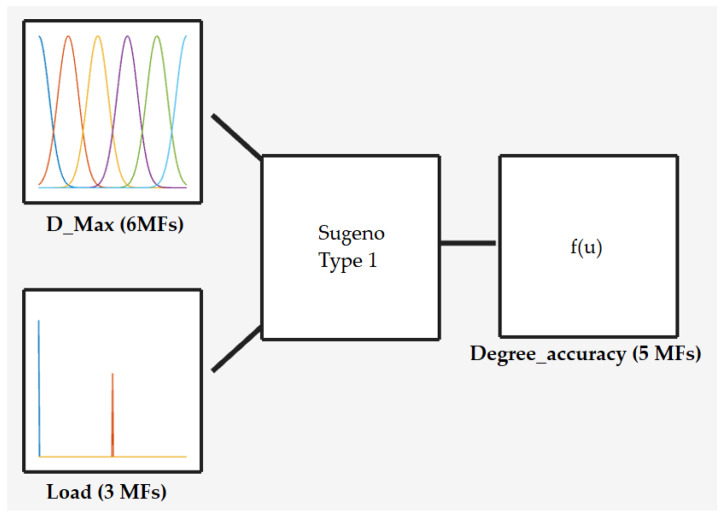
Fuzzy Inference System model.

**Figure 4 jfmk-10-00084-f004:**
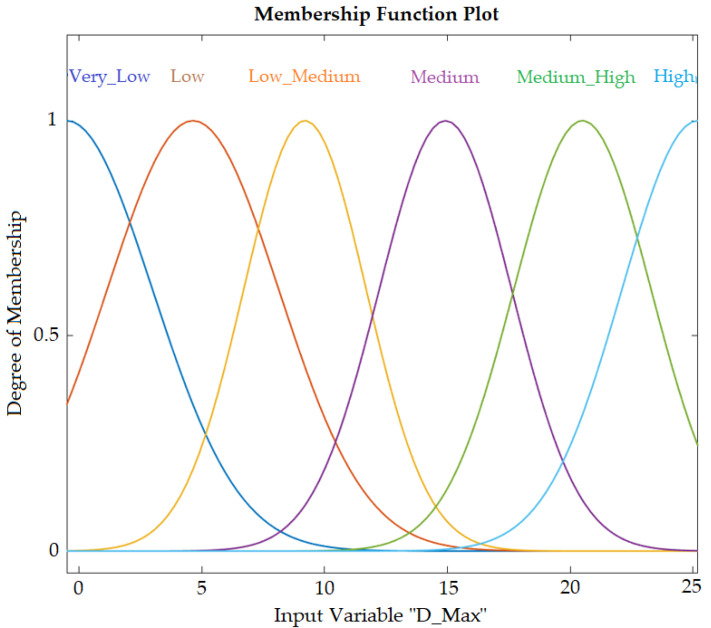
Variation in the values of the absolute deviations at the system input.

**Figure 5 jfmk-10-00084-f005:**
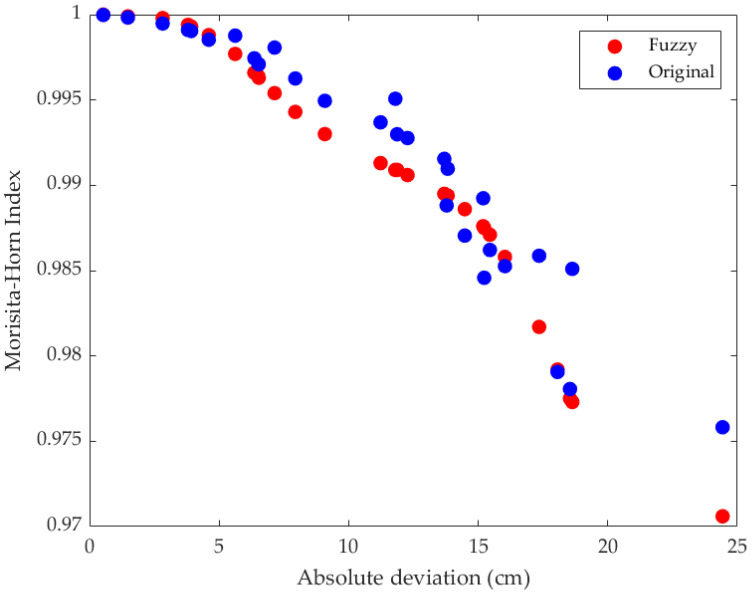
Comparison between original Morisita–Horn data and data obtained from the fuzzy logic model.

**Figure 6 jfmk-10-00084-f006:**
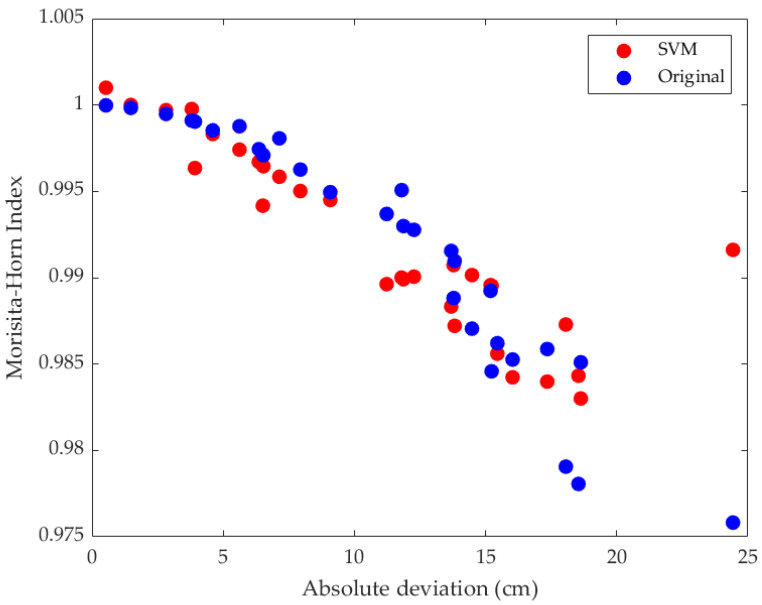
Comparison between original Morisita–Horn data and data obtained by the coarse Gaussian SVM model.

**Table 1 jfmk-10-00084-t001:** Maximum values of absolute horizontal deviations for each volunteer/load.

	Absolute Horizontal Deviation Values (cm)									
Exec.	V01	V02	V03	V04	V05	V06	V07	V08	V09	V10
**Ex1_25**	4.20	10.41	0.88	4.56	3.90	6.71	9.23	4.33	0.55	4.44
**Ex2_25**	4.07	4.69	5.03	3.82	0.97	2.48	2.62	0.13	4.27	3.78
**Ex3_25**	5.25	0.98	4.10	5.32	1.35	1.82	0.5	2.73	7.79	4.18
**Ex1_50**	6.12	4.58	2.47	4.57	5.61	0.15	4.38	1.24	1.45	2.99
**Ex2_50**	5.42	3.58	0.89	4.98	4.03	10.15	1.39	3.03	0.44	5.63
**Ex3_50**	7.55	2.30	0.69	7.63	6.86	4.35	3.63	0.93	1.94	4.05

Ex1_25, execution one at 25% of the subject’s body weight; Ex2_25, execution two at 25% of the subject’s body weight; Ex3_25, execution three at 25% of the subject’s body weight; Ex1_50, execution one at 50% of the subject’s body weight; Ex2_50, execution two at 50% of the subject’s body weight; Ex3_50, execution three at 50% of the subject’s body weight.

**Table 2 jfmk-10-00084-t002:** Maximum values of absolute vertical deviations for each volunteer/load.

	Values of Absolute Vertical Deviations (cm)									
Exec.	V01	V02	V03	V04	V05	V06	V07	V08	V09	V10
**Ex1_25**	1.29	5.61	2.11	0.87	4.25	10.88	18.32	6.03	5.31	10.00
**Ex2_25**	0.64	5.02	6.27	4.03	3.11	4.43	7.59	1.49	12.01	7.63
**Ex3_25**	0.68	9.27	5.82	6.31	2.58	3.95	3.01	3.21	9.44	5.03
**Ex1_50**	7.14	2.82	6.53	15.45	6.51	24.43	15.19	7.94	17.35	15.23
**Ex2_50**	5.62	6.36	1.48	16.03	4.60	13.69	12.27	3.92	11.87	18.06
**Ex3_50**	11.8	9.08	0.53	13.78	14.48	18.63	13.82	3.80	11.23	18.54

Ex1_25, execution one at 25% of the subject’s body weight; Ex2_25, execution two at 25% of the subject’s body weight; Ex3_25, execution three at 25% of the subject’s body weight; Ex1_50, execution one at 50% of the subject’s body weight; Ex2_50, execution two at 50% of the subject’s body weight; Ex3_50, execution three at 50% of the subject’s body weight.

**Table 3 jfmk-10-00084-t003:** Morisita–Horn/Volunteer Indices (vertical).

	Morisita–Horn/Volunteer Indices (Vertical)									
Exec.	V01	V02	V03	V04	V05	V06	V07	V08	V09	V10
**Ex1_50**	0.99807	0.99948	0.99709	0.98621	0.99712	0.97582	0.98924	0.99626	0.98587	0.98458
**Ex2_50**	0.99877	0.99744	0.99983	0.98526	0.99853	0.99155	0.99277	0.99904	0.99299	0.97906
**Ex3_50**	0.99507	0.99495	0.99998	0.98882	0.98705	0.98510	0.99097	0.99910	0.99369	0.97806

**Table 4 jfmk-10-00084-t004:** RMSE and R-squared values for the different SVM models.

	Linear	Quadratic	Cubic	Fine Gaussian	Medium Gaussian	Coarse Gaussian
**RMSE**	0.0038507	0.0071847	0.0053830	0.0041965	0.0040772	0.0031715
**R-squared**	0.72	0.02	0.45	0.66	0.68	0.81

RMSE, root mean square error.

## Data Availability

The data that support the findings of this study are available from the last author (dalton.pessoa-filho@unesp.br) upon reasonable request.

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
