# Peer review of "Similarity Index Values in Fuzzy Logic and the Support Vector Machine Method Applied to the Identification of Changes in Movement Patterns During Biceps-Curl Weight-Lifting Exercise"

_jfmk, 2025, doi:10.3390/jfmk10010084_

Round 1

Reviewer 1 Report

Comments and Suggestions for Authors

Review of Similarity Index Values in Fuzzy Logic and Support Vector Machine Applied to the Identification of Changes in Movement 3 Patterns during Biceps-Curl Weight-Lifting Exercise

I congratulate authors for their great work. I want to point out some issues that I found and might help improving your paper.

1.       Why there is a sentence before Backrgound in the abstract? Structure should be Abstract

Background: text starts here.

2.       Line 143-150 – you mixed aim with method description here. I would stick to stating aim only, maybe with hypothesis.

3.       2.1 is not later repeated with more details. I do not think this is necessary and you can start with participants.

4.       You are giving us information about minimal experience of participants. Can we get mean as for the rest of participants data? At that point reader might starts to wonder how experience could affect results.

5.       I am surprised that you go for resolution instead of fps with camera. Using enhancement of contracts makes this method reliable as you definitely know tracking problem in kinovea and blurring of fast movement in 30 fps. Still, I do not think it was that hard to record in 60 fps or get phone which can do that. S9 is rather old now.

6.       In 2.3 and 2.4 you are describing tracking method and capturing the data. I have hard time finding what was reference distance, as I assume you had to calibrate to get the results in cm as is later stated in the tables.

7.       2.5 is not clear. What was a method to write down their assessments of performance?

8.       I do not think  you need to repeat so many times that you did this all in Matlab.

9.       At results you mention those are results based on analysis of upward movement? Is it correct? Article is about quality of movement of upward phase of biceps curl? Or I get it wrong? Excentric phase and lowering part of the movement is important for movement control.

10.  305-310 still do not know how was this assessment “extracted” from masters.

11.  In discussion you are clearly more interested in models itself, than its usefulness. You are telling readers what to to with your findings, how to possibly apply/deploy this solution to a training or if someone done something similar with the same ML methods.

Overally, I  do not fully understand methodology of experts assessment and you should focus more on practical use of those findings.

Author Response

Thank you for revising this manuscript and for the comments. Please, find the responses in the file attached to this message. The authors appreciated all comments and hope to have addressed each one accordingly.

Reviewer 2 Report

Comments and Suggestions for Authors

The authors present an interesting pilot study of the use of FL algorithm to evaluate bicep curl technique in 10 participants. This is an interesting paper that will require in-depth review by a specialist that can critically assess the development of the algorithm, the equations used and the data analysis more completely than this reviewer is comfortable doing. Some general questions this reviewer has in respect to the manuscript proposal are as follows:

To ensure interater reliability, how were the two experts that provided qualitative analyses trained for standardized assessment?

How were participants trained to standardize their bicep curl technique to ensure uniform and reliable performance?

With respect to the two expert assessors, how was their qualitative assessment reporting standardized?

The above represent major methodological issues that require resolution.

Thank you for the opportunity to review this submission. I look forward to reading the revision.

Author Response

(The authors gave the same response as above.)
